# Dynamic Voltage Restorer Based on Integrated Energy Optimal Compensation

Shengwen Li [1], Xiao Chang [1], Haoqing Wang [2,*], Ning Liu [2], Yizhao Liu [1], Yang Yang [1] and Qing Duan [2]

1   Electric Power Research Institute of State Grid Shanxi Electric Power Company, Taiyuan 030001, China
2   China Electric Power Research Institute, Beijing 100192, China
*   Correspondence: wanghaoqing@epri.sgcc.com.cn

**Abstract:** This paper presents an advanced dynamic voltage restorer with integrated energy optimal compensation (EOC-DVR), which combines the advantages of pre-sag compensation and energy minimization compensation. The performance parameters of the EOC-DVR, such as the compensatory capability, loss of energy, and duration of compensation, can be balanced by this optimal compensation strategy. Additionally, the impact on the load is further reduced by the proposed method. By analyzing the working process of optimal comprehensive energy, the compensation limit of the strategy is discussed. After that, both the voltage and active power of the system for compensating are obtained by analyzing the compensation characteristics of the strategy. Therefore, the relations of the formula and curve between the related physical quantities can be obtained. Finally, simulation results based on MATLAB/Simulink are provided to verify the effectiveness of proposed compensation method.

**Keywords:** dynamic voltage restorer; integrated energy optimal compensation; energy minimized compensation; pre-sag compensation

## 1. Introduction

With the increase in power demand, power users demand high standards of power quality, such as harmonics, voltage sag, three-phase unbalance, etc. [1,2]. Voltage sag has become an important index affecting power quality in recent years [3]. When voltage sag occurs, it will lead to downtime risks for some sensitive equipment. Therefore, it is necessary to consider reasonable and effective methods to compensate for the voltage sag [4,5]. A dynamic voltage restorer (DVR) reduces the effect of voltage sag on the load by injecting a compensating voltage with controllable amplitude and phase. This is the most economical and effective method to restore the dropped voltage [6].

Voltage sag is made up of sag in amplitude and a jump in phase angle. An in-phase compensation method with the characteristics of simple control, fast compensation speed, and large voltage compensation range is proposed to solve this problem [7–9], which means that when the grid voltage sags, the dynamic voltage restorer only compensates the amplitude of the sags, but not the jump phase. Therefore, the amplitude of the compensated voltage is the same as the voltage before the sags, but the phase is different. However, the compensation effect of this strategy is optimal only when the voltage amplitude decreases and the phase does not jump. J. D. Barros et al. [10] and Li B H et al. [11] presented a pre-sag compensation method by which the amplitude and phase angle can be solved simultaneously. However, the demand for active power output is not satisfactory. C. N. Ho et al. [12] and Choi S et al. [13] presented an energy-minimized compensation whose most ideal state of minimum energy compensation is to make the phase of compensation voltage perpendicular to the load current and the compensation effect of the minimum energy compensation strategy is mainly determined by the degree of voltage sag. Secondly, since the minimum energy compensation always seeks to minimize the output active power,

sometimes, the compensated voltage will cause phase jump, which will affect the normal operation of the load. This method can realize reactive power compensation when the voltage sag is shallow. However, when the voltage sag is deep, the injected voltage will be relatively larger in amplitude because it brings the phase jumps artificially. Hence, this method is not appropriate for critical sensitive loads. D. M. Vilathgamuwa et al. [14] presented an improved energy-minimized compensation method, which is suitable for the constant power angle load but not sensitive loads. H. K. Al-Hadidi et al. [15,16] presented an improved energy-minimized compensation with a new topology. However, this solution requires a bulky and costly inductor to meet the high-power and low-frequency requirements. A time-optimized compensation strategy is presented in papers [17,18], which extends the duration of compensation to a certain extent. However, the calculation of this strategy is large and complicated, which is not convenient for engineering applications. The compound compensation strategy for DVR is proposed, as mentioned in the paper [19,20], to combine the advantages of both the pre-sag compensation and the in-phase compensation. In the power supply system of the renewable energy networks, DVR can achieve high-quality power supply for the renewable energy networks. This will benefit the development of renewable energy networks.

An integrated energy optimal compensation, named EOC-DVR, is proposed in this paper. This compensation strategy makes use of the advantage that the minimum energy compensation can realize zero active power compensation when the voltage drop is shallow, and the full voltage compensation can provide the longest compensation time when the voltage drop is deep. The optimal time for switching between the two basic compensation strategies is obtained. Specific analysis is made on the energy loss in the compensation process. In addition, when the required compensation voltage of the power grid exceeds the maximum compensation voltage output by the DVR, considering the compensation strategy under the limit compensation voltage constraint, it ensures the normal operation of the power grid and solves the problem that DVR has limited output compensation voltage capability when the voltage sag is deep. In shallow voltage drop situations, the energy-minimized compensation plays the dominant role. Therefore, the energy loss in compensation is minimized. In deep voltage drop situations, the pre-sag compensation plays the dominant role. Hence, the duration time in compensation is extended. The integrated energy optimal compensation strategy also involves the special case that the voltage required by the grid is beyond the maximum voltage capacity. In addition, the compensation voltage is under the limit. Thus, the stable operation of the power grid is ensured.

The rest of this paper is organized as follows. In Section 2, the topology of the EOC-DVR is presented. The working principle of a traditional compensation strategy is introduced. In Section 3, the theoretical and numerical analysis of the integrated energy optimal compensation is carried out. The best time to switch between compensation strategies is obtained. In Section 4, the compensation characteristics of energy minimized compensation and pre-sag compensation are analyzed in detail. Simulation results are shown in Section 5. Finally, Section 6 concludes this paper.

## 2. The Topology and Fundamental of EOC-DVR

### 2.1. The Topology of EOC-DVR

The proposed EOC-DVR is shown in Figure 1. Under the normal operation of the power grid, the bypass switch turns off to enable EOC-DVR to enter the bypass mode. When the line voltage drops, the bypass switch will turn on. The compensation voltage is generated by the inverter. The PWM wave generated by the controller is transmitted to the inverter as a control signal. Voltage is compensated into the power line by a series transformer, then restored to the normal level.

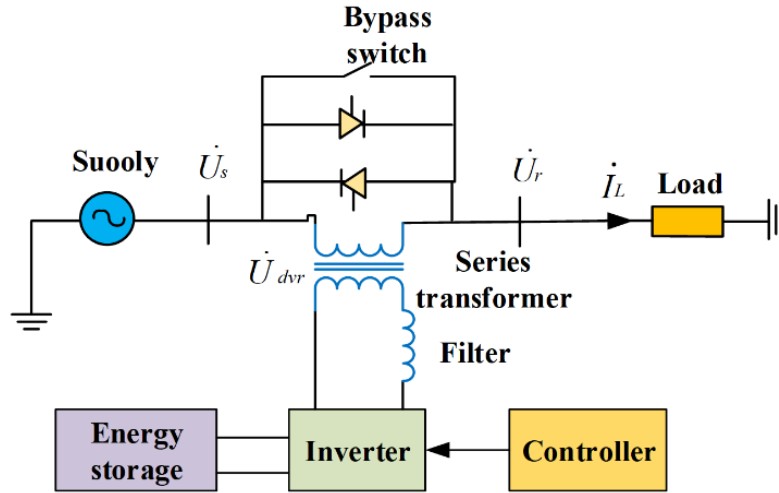

**Figure 1.** Principle diagram of the proposed EOC-DVR.

The compensation voltage of EOC-DVR is as follows:

$$U_{dvr}\angle\varphi_{dvr} = U_r\angle 0 - U_s\angle\varphi_s \tag{1}$$

where $U_s$ is the amplitude of the voltage sag, $\varphi_s$ is the phase angle of the voltage sag, $\varphi_r$ and $U_r$ are the references to the load voltage, and $\varphi_{DVR}$ and $U_{DVR}$ are the output of the EOC-DVR.

### 2.2. Fundamental of EOC-DVR Compensation

When the voltage drops, the EOC-DVR needs to compensate for the voltage and exchange energy with the system. The EOC-DVR achieves the longest voltage sag compensation time while reducing the capacity of the accumulator unit. Meanwhile, the voltage sag can be controlled to reduce the energy exchange by selecting an appropriate compensation strategy. There are three compensation methods for switching.

### 2.2.1. In-Phase Compensation

This method is to directly compensate the voltage amplitude to the initial level, as shown in Figure 2. In-phase compensation can quickly compensate for the voltage amplitude with minimum output voltage but does not consider the voltage phase jump. This method is not conducive to the normal operation of the sensitive load.

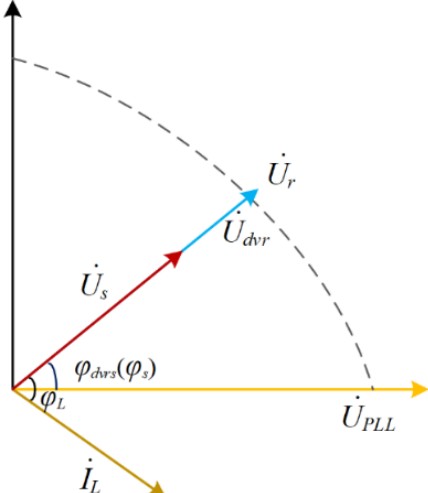

**Figure 2.** In-phase compensation phasor diagram.

In Figure 2, $I_L$ and $\varphi_L$ are the load current and power factor, respectively. $U_{PLL}$ is the reference phasor of the voltage.

According to Figure 2, the voltage and amplitude of the compensation output for EOC-DVR are:

$$U_{dvr} = U_r - U_s \tag{2}$$

$$\varphi_{dvr} = \varphi_s \tag{3}$$

The output active power can be obtained as:

$$P_{dvr} = (U_r - U_s)I_L \cos \varphi_L \tag{4}$$

### 2.2.2. The Pre-Sag Compensation

The EOC-DVR injects compensation voltage into the grid to keep the amplitude and phase of the voltage in its initial form before the drop, as shown in Figure 3. Further, it can ensure the continuous operation of the grid voltage and reduce the impact of voltage sag. However, this method requires a high compensation range and a big capacity for energy storage with low effectiveness.

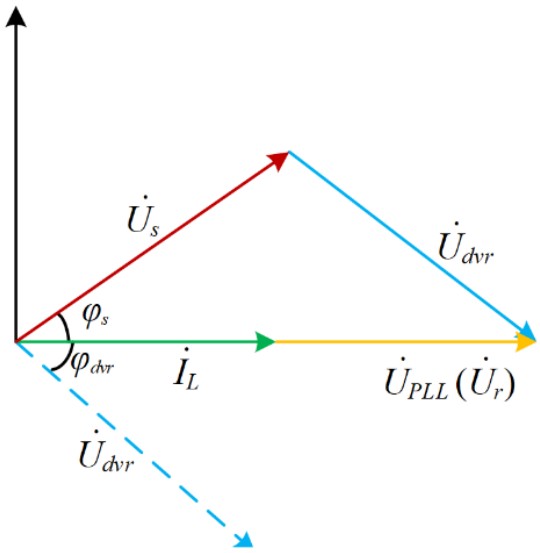

**Figure 3.** Compensation phasor diagram of pre-sag.

Figure 3 shows the compensation voltage of pre-sag compensation:

$$U_{dvr} = \sqrt{U_s^2 + U_r^2 - 2U_sU_r \cos \varphi_s} \tag{5}$$

$$\varphi_{dvr} = -\arccos(\frac{U_{dvr}^2 + U_r^2 - U_s^2}{2U_{dvr}U_r}) \tag{6}$$

The output of active power $P_{EOC-DVR}$ is expressed as:

$$P_{dvr} = U_rI_L \cos \varphi_L - U_sI_L \cos(\varphi_L - \varphi_s) \tag{7}$$

### 2.2.3. Energy Minimized Compensation

Energy minimization compensation is shown in Figure 4. It is made perpendicular to the load current by adjusting the phase of the EOC-DVR compensation voltage. When the load voltage amplitude remains unchanged, the EOC-DVR reduces active power output and energy loss to extend the duration of grid voltage compensation while the load voltage amplitude remains unchanged. When voltage dips deeply, EOC-DVR ensures that the load voltage returns to the original state before the dip by providing active power. However,

at this time, EOC-DVR needs to output a relatively large voltage amplitude, and the increase in the voltage amplitude also means that more energy is consumed, so the energy minimized compensation cannot reach the purpose of extending the voltage compensation.

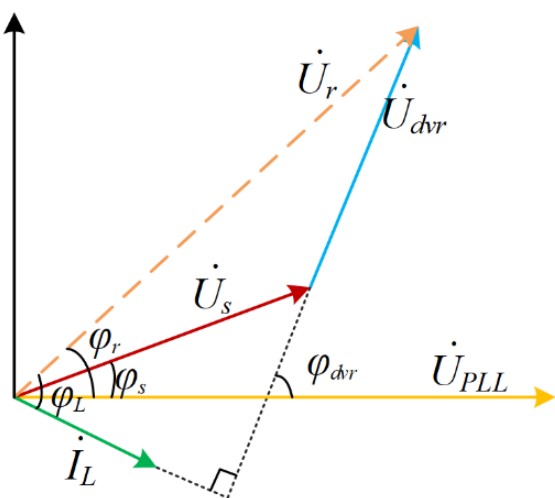

**Figure 4.** Phasor diagram of the energy minimized compensation.

In Figure 4, the single-phase system active power under energy minimization compensation is as follows:

$$P_s = U_r I_L \cos(\varphi_L - \varphi_r + \varphi_s) \tag{8}$$

The load's active power requirements are as follows:

$$P_L = U_r I_L \cos \varphi_L \tag{9}$$

The voltage and power of EOC-DVR are expressed as:

$$U_{dvr} = \sqrt{U_r^2 + U_s^2 - 2U_s U_r \cos(\varphi_r - \varphi_s)} \tag{10}$$

$$P_{dvr} = U_r I_L \cos \varphi_L - U_s I_L \cos(\varphi_L - \varphi_r + \varphi_s) \tag{11}$$

## 3. Integrated Energy Optimal Compensation

### 3.1. Derivation of Integrated Energy Optimal Compensation

EOC-DVR can obtain optimal energy compensation. The lower the EOC-DVR output active power, the slower the energy consumption will be, and the purpose of extending the EOC-DVR compensation duration will be achieved. The energy-minimized compensation is presented based on this principle. From the above analysis, it can be concluded that when $U_s \geq U_r \cos \varphi_L$, the energy-minimized compensation can realize zero active work compensation, and at this point, if the influence of loss is ignored, the compensation time can be infinite. If the voltage sag is relatively deep, i.e., $U_s < U_r \cos \varphi_L$. At that time, the active power is already minimum, but the voltage is still high. The energy-minimized compensation cannot ensure that the most continuous voltage compensation can be provided under any circumstances, because the compensation duration is associated with the power and voltage together.

The compensation time is expressed as:

$$T = C * \frac{U_{dc}^2 - \left(\frac{\sqrt{2}U_{dvr}}{MN}\right)^2}{2P_{dvr}} \tag{12}$$

where $U_{dc}$ is the DC voltage, $C$ is the value of bus capacitance, $M$ is the maximum modulation ratio of the PWM, and $N$ is the transformer variation ratio.

Substitute Equations (2) and (4) into Equation (12), the time available for in-phase compensation as follows:

$$T_1 = C * \frac{U_{dc}^2 - K(U_r - U_s)^2}{2(U_r - U_s)I_L \cos \varphi_L} \tag{13}$$

Substitute Equations (5) and (7) into Equation (12), the time available for pre-sag compensation is:

$$T_2 = C * \frac{U_{dc}^2 - K(U_s^2 + U_r^2 - 2U_s U_r \cos \varphi_s)}{2[A - U_s I_L \cos(\varphi_L - \varphi_s)]} \tag{14}$$

where $A = U_r I_L \cos \varphi_L$, $K = \frac{2}{(MN)^2}$.

Substituting Equations (10) and (11) into Equation (12), the time available for energy minimized compensation can be calculated as:

$$T_2 = C * \frac{U_{dc}^2 - K[U_s^2 + U_r^2 - 2U_s U_r \cos(\varphi_r - \varphi_s)]}{2[A - U_s I_L \cos(\varphi_L - \varphi_r + \varphi_s)]} \tag{15}$$

When $U_s \geq U_r \cos \varphi_L$, the energy-minimized compensation can realize the infinite length of compensation duration. Figure 5 shows the compensation duration of the three compensation strategies when the voltage of the grid jump angle is 10° and the load power angle is 30°. Figure 5 shows that the energy-minimized compensation provides the longest compensation duration only when the amplitude of voltage sag is small, and the pre-sag compensation provides the longstanding compensation duration when it is deeper.

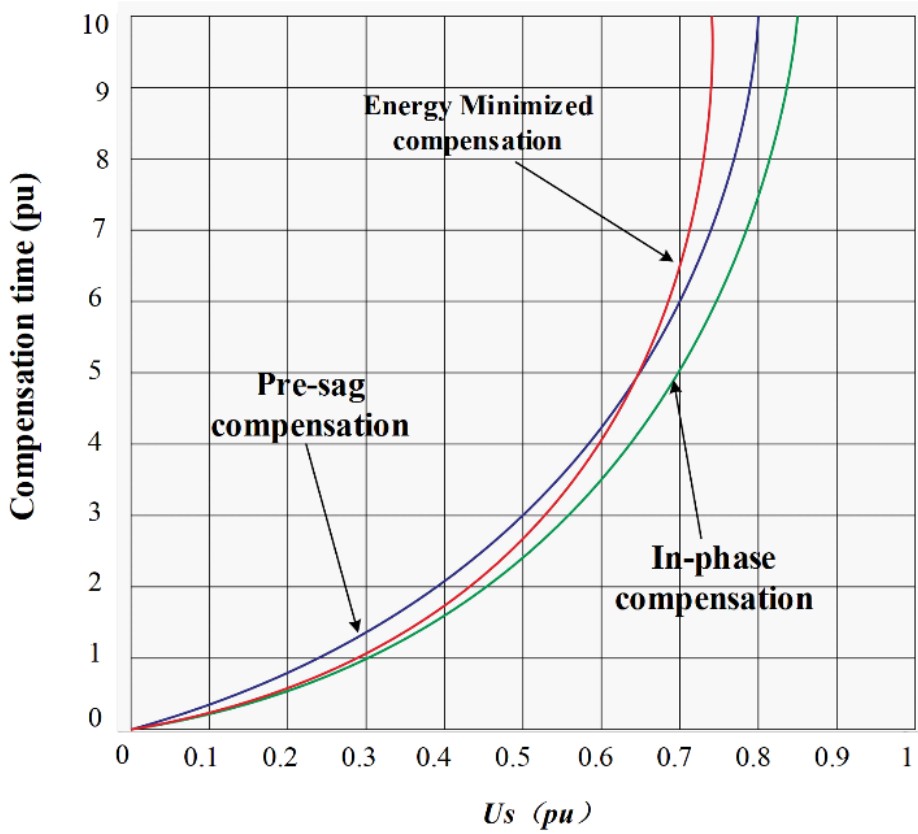

**Figure 5.** Compensation duration for three compensation strategies.

After the above analysis of the compensation duration of the three commonly used compensation strategies and the analysis of the advantages and disadvantages of the three compensation strategies in Section 2.2 above, based on the advantages of energy-minimized compensation and pre-sag compensation, this paper proposes an integrated energy optimal

compensation strategy. When $U_s \geq U_r \cos \varphi_L$, the energy-minimized compensation is used to realize the zero-active output of the EOC-DVR device; when $U_s < U_r \cos \varphi_L$, pre-sag compensation is used to increase the time of voltage compensation by EOC-DVR. This compensation strategy can realize the compensation of voltage phase angle jump and the extension of voltage compensation duration. At the same time, this strategy also considers the special case when the required voltage of the power network exceeds the maximum voltage provided by the EOC-DVR and considers the compensation strategy under the limit voltage constraint. The working mode of integrated energy optimal compensation is shown below.

$$
\left\{
\begin{array}{l}
\text{The minimum energy mode, } U_s \geq U_r \cos \varphi_L \\
\text{The full voltage compensation mode, } U_s < U_r \cos \varphi_L \\
\text{The limit voltage compensation mode, } U_{dvr} \geq U_{dvr\,\max}
\end{array}
\right.
$$

### 3.2. Principle of Integrated Energy Optimal Compensation

From Figure 4, it can be deduced that when $U_s \geq U_r \cos \varphi_L$, the energy minimization compensation with shallow voltage sag can always be used to find when the load current and compensation voltage are perpendicular to each other. Then, the power injected by the EOC-DVR into the grid is equal to 0. If $P_{EOC-DVR} < 0$, the reversal phenomenon of EOC-DVR absorbs power from the grid and causes DC voltage instability, affecting the stable operation of EOC-DVR.

According to Equation (11), the compensation voltage of EOC-DVR is related to the phase angle $U_r$ of active power output. Assuming $P_{EOC-DVR} = 0$, the compensated reference voltage phase is obtained according to the following Equation (11):

$$
\varphi_r = \varphi_L + \varphi_s - \arccos\left(\frac{U_r \cos \varphi_L}{U_s}\right) \tag{16}
$$

To sum up, the compensation voltage of EOC-DVR output is expressed based on Equatiopn (16).

The phase angle of the output voltage of EOC-DVR can be expressed as

$$
\varphi_{dvr} = \frac{\pi}{2} - (\varphi_L - \varphi_r) \tag{17}
$$

According to Equations (1) and (17), the voltage phase angle of the grid side can be obtained as

$$
\varphi_s = \arccos\left[\frac{U_r}{\sqrt{U_s^2(1 + \tan^2 \varphi_L)}}\right] \tag{18}
$$

According to Equation (16), when $U_s \geq U_r \cos \varphi_L$, the formula is true. If the voltage sag is too deep, $U_s < U_r \cos \varphi_L$, the EOC-DVR device will output active power.

When $U_s < U_r \cos \varphi_L$, as just shown in Figure 6, the energy minimization compensation continues to be used, and a relatively large voltage amplitude needs to be output at this time. Literature has shown that the increase in the amplitude of compensation voltage will also consume more energy, resulting in a reduction in compensation time. When the voltage sag is relatively deep, the compensation duration of pre-sag compensation is extended. At present, it is the best way to adopt pre-sag compensation. If the amplitude is shallow, energy minimization compensation will be adopted. If the amplitude is deep, pre-sag compensation is suited for use.

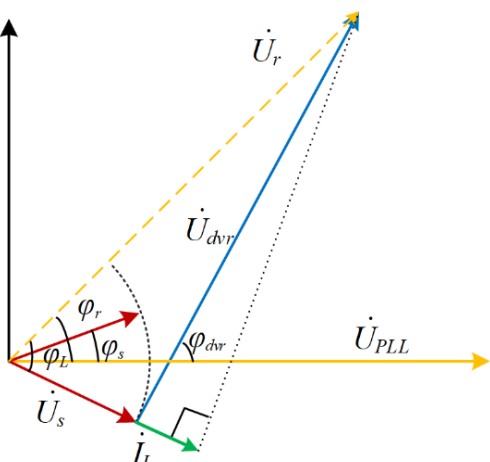

**Figure 6.** Minimized compensation phasor diagram of non-zero active energy.

### 3.3. Limit Voltage Compensation for Integrated Energy Optimal Compensation

When the limit compensation voltage of the EOC-DVR change, which may cause $U_{EOC-DVR}$, calculated by Equation (5), to be greater than the limit compensation voltage of the EOC-DVR. Therefore, the integrated energy optimal compensation switches to pre-sag compensation mode, according to the analysis in Section 2.1, when the voltage sag is relatively deep. Thus, the limit voltage compensation is realized based on the pre-sag compensation mode.

When voltage compensation is limited, the minimum phase jump of load side voltage compensation should be set as the goal, according to the rule of pre-sag compensation. Based on this goal, this paper proposes to adjust the amplitude of reference voltage within the allowable range of load voltage and reduce the amplitude of reference voltage. The output limit voltage of the EOC-DVR is taken as the compensation voltage (i.e., $U_{EOC-DVR} = U_{EOC-DVR\ \text{max}}$), and the load voltage after compensation determines the load reference voltage according to no phase change. At this time, the maximum value that the EOC-DVR can compensate for is based on ensuring that the load voltage phase does not change. The adjustment is shown in Figure 7. The small circular arc is the compensation range of the EOC-DVR limit voltage. Currently, the load reference voltage can be obtained as:

$$U'_r = U_s \sin \varphi_s + \sqrt{U_{dvr\text{max}}^2 - (U_s \sin \varphi_s)^2} \tag{19}$$

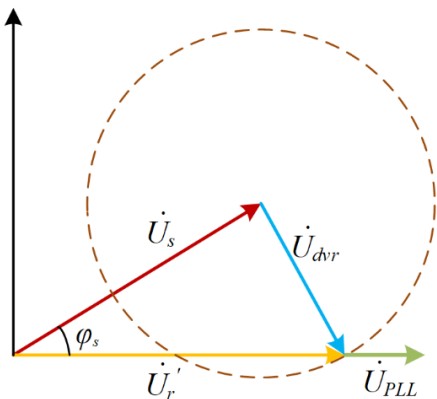

**Figure 7.** Limit voltage compensation strategy 1.

If the load reference voltage $U'_r$ determined by the above method exceeds the allowable range of load voltage, then the output limit voltage of the EOC-DVR will be used as the compensation voltage. In this condition, the maximum allowable voltage on the load side

$U_{L\,\max}$ is used as the reference voltage. Furthermore, $U'_r$ and $\varphi_r$ need to reduce as small as possible.

As shown in Figure 8, the small circular arc is the scope of the EOC-DVR limit compensation voltage, and the large arc is the range of the load reference voltage. In Figure 7, two compensation reference voltage phasors can be obtained at this time, and the smaller phase is selected as the compensation voltage, which can ensure the minimum phase change of the load after compensation.

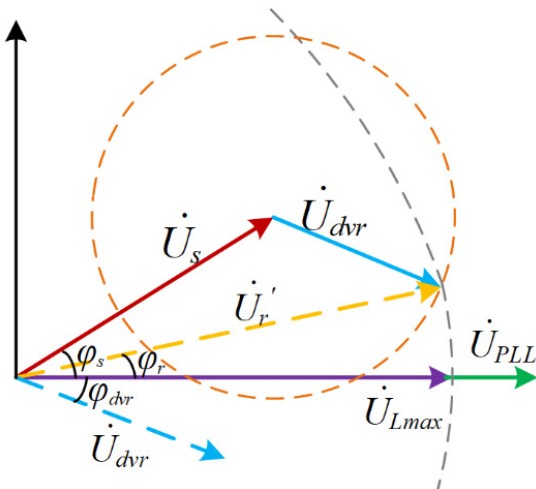

**Figure 8.** Limit voltage compensation strategy 2.

## 4. Analysis of Compensation Characteristics

To reflect the steady-state compensation characteristics of the integrated energy optimal compensation more intuitively, the standard unit value is adopted as the per-unit value of each quantity. $U_r$ is defined as the reference voltage, valuing 1 p.u. $I_L$ is defined as the reference current, valuing 1 p.u. Additionally, the per-unit value of the grid voltage is $U_s^*$. The per-unit value of the compensation voltage output by EOC-DVR is $U_{EOC-DVR}^*$.

### 4.1. Energy Minimized Compensation Model

The EOC-DVR works in the energy-minimized compensation mode, i.e., $U_s \geq U_r \cos \varphi_L$, the grid voltage and the PF of the load should meet $U_s^* > \cos \varphi_L$. Then, the power output from EOC-DVR is 0. According to Equation (10), the compensation voltage per-unit value $U_{EOC-DVR}^*$ can be defined as:

$$U_{dvr}^* = \sin \varphi_L + \sqrt{U_s^2 - \cos^2 \varphi_L} \tag{20}$$

According to Equation (20), the output voltage of the energy-minimized compensation mode is graphically represented, as shown in Figure 9. In the figure, the x-coordinate is the grid voltage per-unit value $U_s^*$, the y-coordinate is the PF of load $\cos \varphi_L$, and the vertical coordinate is the compensation voltage amplitude per-unit value $U_{EOC-DVR}^*$.

### 4.2. Pre-Sag Compensation Mode

The EOC-DVR works in the pre-sag compensation mode, i.e., $U_s < U_r \cos \varphi_L$. From the analysis of the phasor diagram in Figure 3, the grid voltage and the power factor of the load should meet $U_s^* < \cos \varphi_L$. At this point, according to Equation (5), the compensation per-unit value $U_{EOC-DVR}^*$ of the EOC-DVR output can be calculated as:

$$U_{dvr}^* = \sqrt{1 + U_s^{*2} - 2U_s^* \cos \varphi_s} \tag{21}$$

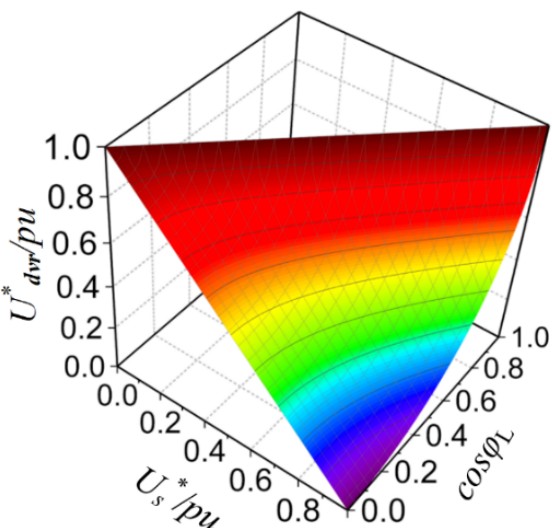

**Figure 9.** Quantitative analysis of compensation voltage for integrated energy compensation with $U_s < U_r \cos \varphi_L$.

According to Equation (7), the per-unit value $P_{dvr}^*$ of the active power output from the EOC-DVR is:

$$P_{dvr}^* = 1 - U_s^* \cos \varphi_s \tag{22}$$

According to Equations (21) and (22), a graphical representation of the voltage and power in the pre-sag compensation mode is obtained. The vertical coordinates are the compensated voltage amplitude per-unit value $U_{EOC-DVR}^*$ or the output active power per-unit value $P_{EOC-DVR}^*$. The abscissa is the grid voltage per-unit value $U_s^*$, and the ordinate is the power factor $\cos \varphi_s$ of the power supply, as shown in Figure 10.

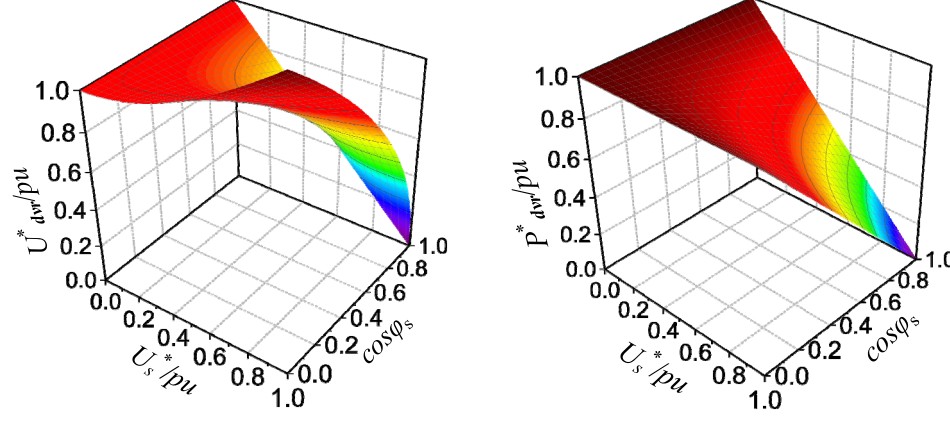

(**a**) Compensating voltage amplitude          (**b**) Output active power

**Figure 10.** Quantitative analysis of optimal energy compensation with $U_s < U_r \cos \varphi_L$.

When the EOC-DVR works in the limit compensation mode, the compensation voltage is always the limit compensation voltage of the EOC-DVR, and the voltage amplitude is constant, which is not represented graphically here.

It can be seen from the diagram of compensation characteristic analysis in Figures 9 and 10 that the compensation voltage and output active power of the EOC-DVR is related to the grid voltage sag and load PF. According to the voltage and load PF in voltage sag time, both the compensation voltage and output power of the EOC-DVR can be obtained.

## 5. Simulation Verification

When the proposed control strategy is actually implemented, the compound control is actually the combination of feed-forward control and feedback control. In the feedback control, the output voltage of the DVR changes with the difference between the load voltage and the rated voltage of the power grid. When the voltage drop of the power grid is deep, the difference between the two is large, and the adjustment of the closed-loop control will be more frequent, which puts forward higher requirements for the output capacity of the DVR. After adding the feed-forward control of the instantaneous value of the grid voltage on the basis of the feedback control, the feed-forward control will first pre-compensate the sag voltage. Therefore, to validate the proposed compensation method, a simulation model of the integrated energy optimal compensation based on the EOC-DVR is developed in MATLAB/Simulink software. The key simulation parameters are shown in Table 1.

**Table 1.** Simulation parameters of EOC-DVR.

| Parameter | Value |
|---|---|
| Rated peak voltage on the grid side, $U_{smref}/V$ | 311 |
| Rated frequency of grid side, $f/Hz$ | 50 |
| Dc side rated voltage, $U_{dc}/V$ | 400 |
| Filter inductance, $L/mH$ | 4 |
| Filter capacitor, $C/\mu F$ | 50 |
| Load resistance, $R_o/\Omega$ | 7.7 |
| Load inductance, $L_o/mH$ | 25 |
| Load power factor, $\cos \varphi_L$ | 0.8 |

In the simulation experiment, the energy minimized compensation and the integrated energy optimal compensation are respectively used for simulation analysis. Each group of data is composed of four channel signals. Grid voltage, EOC-DVR compensation voltage, compensated load voltage, and load current are arranged from top to bottom in the figure.

Figure 11 shows the results of DVR in the energy-minimized compensation. Figure 12 shows the results of EOC-DVR in other modes when the integrated energy optimal compensation is adopted. From the simulation results, for integrated energy optimal compensation, no matter which mode it works in, the compensation effect is better than the energy minimized compensation during the voltage sag from 0.3 to 0.5 s. From the compensation voltage $U_{EOC-DVR}$ of the EOC-DVR, we can see that the phase angle of the output voltage amplitude is significantly smaller when the energy minimized compensation is close to 0.5 s, while the compensation voltage is closer to the initial state. However, load voltage $U_r$ has not completely recovered to the level before, and the load voltage is restored while using the integrated energy optimal compensation. As seen from the load current $I_L$, the energy-minimized compensation has caused a certain degree of interference to the working performance of the load, while the integrated energy optimal compensation has less interference to the load so that the load can be restored to the normal working state.

Figure 13 shows the comparison between the energy minimized compensation and the integrated energy optimal compensation's EOC-DVR output active power. During the voltage sag period from 0.3 to 0.5 s, the integrated energy optimal compensation can reduce the energy loss more than the energy minimized compensation. Therefore, the purpose of extending the voltage sag compensation duration is achieved.

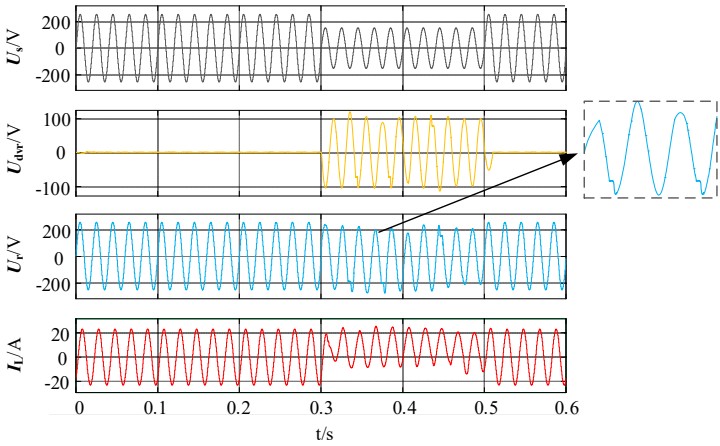

**Figure 11.** Energy minimized compensation simulation results.

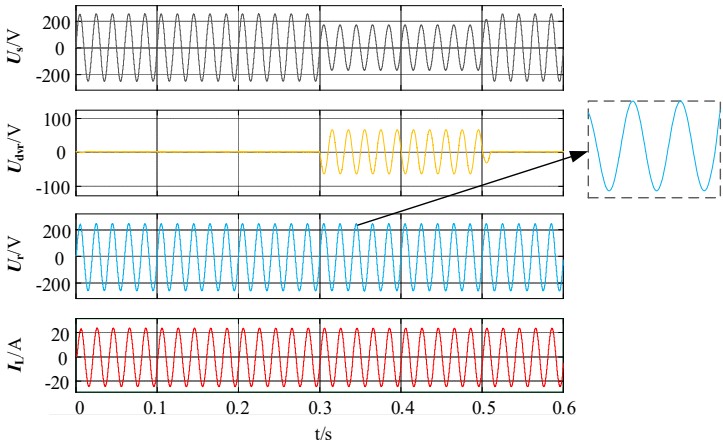

(**a**) Voltage sag by 30%

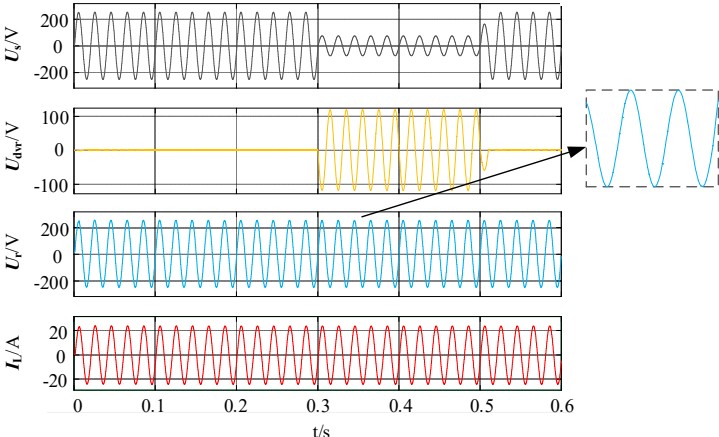

(**b**) Voltage sag by 70%

**Figure 12.** Simulation results of integrated energy optimal compensation.

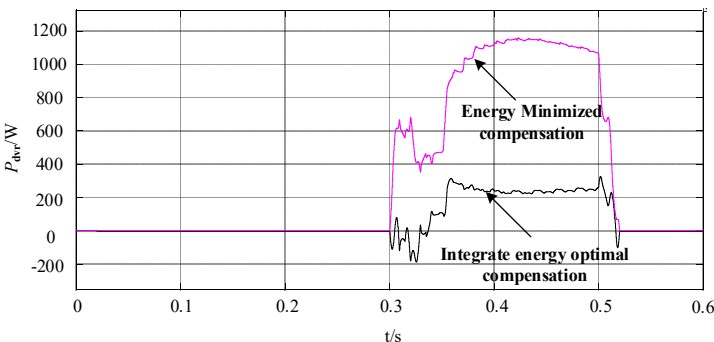

**Figure 13.** Output active power.

To sum up, the integrated energy optimal compensation has a better compensation effect for different degrees of voltage sags, which reduces the active power output and effectively extends the voltage compensation duration. At the same time, the analysis results of the compensation characteristics of the integrated energy optimal compensation strategy are verified, and the feasibility of the compensation strategy is verified.

## 6. Conclusions

Aiming at the voltage compensation problem of voltage sag in the power system, an integrated energy optimal compensation method is proposed. From the theoretical analysis and simulation results, we reached the following conclusions:

(1) The integrated energy optimal compensation strategy fully combines the advantages of two traditional compensation methods. The comprehensive balance of compensatory capability, energy loss, and compensation duration is realized by this optimal compensation method.

(2) In limit compensation mode, the amplitude of compensation voltage is constant. Therefore, EOC-DVR can be more stable when the needed compensation voltage is exceeded.

(3) The simulation results show that the proposed compensation method reduces the energy loss of EOC-DVR and prolongs the voltage sag compensation duration of EOC-DVR.

**Author Contributions:** For research articles, Conceptualization, S.L. and H.W.; methodology, X.C.; software, X.C.; validation, Y.L., Y.Y. and Q.D.; formal analysis, X.C.; investigation, X.C.; resources, X.C.; data curation, X.C.; writing—original draft preparation, X.C. and N.L.; writing—review and editing, X.C.; visualization, X.C.; supervision, X.C.; project administration, X.C.; funding acquisition, S.L. All authors have read and agreed to the published version of the manuscript.

**Funding:** This research was supported by two research projects, including major science and technology projects of Shanxi province (No: 20181102028) and the Science and Technology Project of State Grid Shanxi Electric Power Company (No: 520530200014).

**Conflicts of Interest:** The authors declare no conflict of interest.

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
