# Peer review of "Dynamic Voltage Restorer Based on Integrated Energy Optimal Compensation"

_electronics, doi:10.3390/electronics12030531_

Round 1

Reviewer 1 Report

See attached.

Reviewer 2 Report

Dynamic Voltage Restorer Based on Integrated Energy Optimal Compensation

by Shengwen Li, Xiao Chang, Haoqing Wang, Ning Liu, Yizhao Liu, Yang Yang, Qing Duan

For the Special Issue “IoT Applications for Renewable Energy Management and Control

 Review

This paper presents a strategy for the energy optimal compensation in dynamic voltage restorer (DVR), based on pre-sag compensation and energy minimized compensation. The compensation limit of the strategy is discussed. The simulation model is built on MATLAB/Simulink and verifies the proposed strategy. The simulation results show that when the voltage phase angle and sag are recovered, the output active power of the DVR is reduced and the duration of compensation is extended.

The paper is interesting, well written and complete, with research  hypothesis, development of analytical model and numerical simulation and discussion of results.

Is this reviewer’s opinion that some minor editing will add quality.

The numbering of references must be verified: in the section References there are 19 references, while in the section Introduction (pages 1-2) are mentioned 20 references.

A description of a practical implementation of the proposed strategy will be interesting for the readers.

Finally, the subject of this manuscript is not related to the topic of the Special Issue “IoT Applications for Renewable Energy Management and Control”. The authors should provide a paragraph to justify the submission, by showing the implementation of the proposed strategy in renewable energy networks and with Internet of Things.
